# A Multicenter Analysis of the Outcome of Cancer Patients with Neutropenia and COVID-19 Optionally Treated with Granulocyte-Colony Stimulating Factor (G-CSF): A Comparative Analysis

**DOI:** 10.3390/cancers13164205

**Published:** 2021-08-20

**Authors:** María Sereno, Ana María Jimenez-Gordo, Javier Baena-Espinar, Carlos Aguado, Xabier Mielgo, Ana Pertejo, Rosa Álvarez-Álvarez, Ana Sánchez, Jose Luis López, Raquel Molina, Ana López-Alfonso, Berta Hernández, Luis Enrique Chiara, Ana Manuela Martín, Ana López-Martín, Miriam Dorta, Ana Collazo-Lorduy, Enrique Casado, Ana Ramirez de Molina, Gonzalo Colmenarejo

**Affiliations:** 1Infanta Sofía University Hospital, 28702 Madrid, Spain; ajgordo@salud.madrid.org (A.M.J.-G.); enriquecasado@salud.madrid.org (E.C.); 2IMDEA-Food Institute, CEI UAM+CSIC, 28702 Madrid, Spain; aramirez@iib.uam.es (A.R.d.M.); gonzalo.colmenarejo@imdea.org (G.C.); 3Doce de Octubre University Hospital, 28702 Madrid, Spain; javier.baena@salud.madrid.org; 4San Carlos University Hospital, 28040 Madrid, Spain; carlos.aguado@salud.madrid.org; 5Fundación-University Alcorcón Hospital, 28922 Madrid, Spain; xmielgo@salud.madrid.org; 6La Paz University Hospital, 28046 Madrid, Spain; pertejo.ana@salud.madrid.org; 7Gregorio Marañón University Hospital, 28009 Madrid, Spain; rosa.alvarez.al@salud.madrid.org; 8Getafe University Hospital, 28905 Madrid, Spain; anasanpen@salud.madrid.org; 9Príncipe de Asturias University Hospital, 28805 Madrid, Spain; jllopez@salud.madrid.org (J.L.L.); raquel.molina@salud.madrid.org (R.M.); 10Infanta Leonor University Hospital, 28031 Madrid, Spain; ana.lopez2@salud.madrid.org; 11La Princesa University Hospital, 28006 Madrid, Spain; bertahernandezmarin@salud.madrid.org; 12Guadalajara University Hospital, 19001 Castilla La Mancha, Spain; lucho_chara@salud.castillaylamancha.com; 13Fuenlabrada University Hospital, 28942 Madrid, Spain; anamanuela.martin@salud.madrid.org; 14Leganés-Severo Ochoa University Hospital, 28914 Madrid, Spain; ana5lomar@salud.madrid.org; 15Clara Campall CIOCC, 28050 Madrid, Spain; mdorta@hmhospitales.com; 16Puerta de Hierro University Hospital, 28220 Madrid, Spain; anaclorduy@salud.madrid.org

**Keywords:** COVID-19, neutropenia, G-CSF treatment, respiratory failure

## Abstract

**Simple Summary:**

Infections with COVID-19 in neutropenic cancer patients are related to poor outcomes. A G-CSF treatment used in neutropenic cancer with SARS-CoV-2 infections is related to a higher rate of respiratory failure according to progressive and growing evidence. In this small retrospective non-randomized study, we found an association between G-CSF treatment and the parameters predisposing for worse infections with COVID-19 and neutropenia compared with patients not treated with G-CSF. We also found that the number of days on G-CSF treatment was related to a higher risk of mortality in a multivariable analysis among patients treated with G-CSF.

**Abstract:**

Background: Approximately 15% of patients infected by SARS-CoV-2 develop a distress syndrome secondary to a host hyperinflammatory response induced by a cytokine storm. Myelosuppression is associated with a higher risk of infections and mortality. There are data to support methods of management for neutropenia and COVID-19. We present a multicenter experience during the first COVID-19 outbreak in neutropenic cancer patients infected by SARS-CoV-2. Methods: Clinical retrospective data were collected from neutropenic cancer patients with COVID-19. Comorbidities, tumor type, stage, treatment, neutropenia severity, G-CSF, COVID-19 parameters, and mortality were analyzed. A bivariate analysis of the impact on mortality was carried out. Additionally, we performed a multivariable logistic regression to predict respiratory failure and death. Results: Among the 943 cancer patients screened, 83 patients (11.3%) simultaneously had neutropenia and an infection with COVID-19. The lungs (26%) and breasts (22%) were the primary locations affected, and most patients had advanced disease (67%). In the logistic model, as adjusted covariates, sex, age, treatment (palliative vs. curative), tumor type, and the lowest level of neutrophils were used. A significant effect was obtained for the number of days of G-CSF treatment (OR = 1.4, 95% CI [1,1,03,92], *p*-value = 0.01). Conclusions: Our findings suggest that a prolonged G-CSF treatment could be disadvantageous for these cancer patients with infections by COVID-19, with a higher probability of worse outcome.

## 1. Introduction

Coronavirus disease 2019 (COVID-19), caused by severe acute respiratory syndrome coronavirus 2 (SARS-CoV-2), has become a global pandemic. Infections by SARS-CoV-2 can turn into acute respiratory proliferation, which is cleared in most cases by the immune system after 7 to 14 days [1]. However, approximately 15% of patients infected by COVID-19 develop severe lung disease and multiorgan failure, which are major causes of mortality [2]. Two different but overlapping phases can be distinguished in COVID-19: an initial mild response to the virus infection, followed by a severe phase with a host hyperinflammatory response induced by a cytokine storm [3,4]. The development of novel therapies are critical to overcome this pandemic [5]. It is well known that cancer patients are more vulnerable to infections [6]. Jung et al. described a rate of 0.79% SARS-CoV-2 infections in 1,524 cancer patients compared with a rate of 0.37% in the general population of Wuhan [7]. Kim et al. described a much higher mortality in patients with hematologic malignancies and infection with COVID-19 compared with others without this condition (40 vs. 3.6%, *p* < 0.001) [8]. Furthermore, most of the patients with comorbidities such as hypertension, diabetes, obesity, chronic obstructive pulmonary disease (COPD), tobacco consumption, vascular disease, advanced age, and other chronic diseases require frequent visits to the hospital, which is correlated with a higher risk of severe complications in the case of infection with SARS-CoV-2 [9,10].

Chemotherapy is one of the most common treatments in cancer patients. Myelosuppression, more specifically the development of febrile neutropenia, is an undesirable secondary effect in these patients [11]. Soon after the onset of the pandemic, several panels of experts in Spanish, European, and American oncology associations recommended the modification of oncological treatments to induce less neutropenia and to consider expanding the indications of Filgrastim (G-CSF) use to patients with intermediate (10–20%) and higher neutropenia risk. Scarce data have been reported about the effect of G-CSF in patients with cancer and infections by COVID-19. Lymphocyte T-mediated immunity has been implicated in SARS-CoV-2 infections [12]. G-CSF has been associated with a reduction in hospitalization days and a quicker recovery of neutrophil counts, including in older febrile neutropenia patients [13]. The mechanism of action of G-CSF includes the stimulation of both cytokines and neutrophils. This has been associated with lung injury, including adult respiratory distress syndrome (SDRA) [14]. Moreover, neutrophilia and a high neutrophil/lymphocyte ratio (NLR) have been described as bad prognostic factors in patients infected with COVID-19 [15]. Thus, there is uncertainty about the use of G-CSF and its impact in the clinical outcome of cancer patients treated with it [16]. 

To shed some light on this convoluted issue, in this work, we present a multi-center experience in several hospitals in Spain during the first COVID-19 outbreak with a cohort of patients with neutropenia cancer and infected with COVID-19. 

## 2. Materials and Methods

This is a retrospective and observational analysis (without randomized design) and includes patients with neutropenic cancer and simultaneously infected with SARS-CoV-2. We collected cases from 14 hospitals from Madrid and Guadalajara in Spain (H. 12 de Octubre, H. Clínico San Carlos, HF. Alcorcón, H. Infanta Sofía, H. Infanta Leonor, H. Guadalajara, H. Puerta de Hierro, H. Severo Ochoa, H. Alcalá de Henares, H. Getafe, H. La Paz, H. Princesa, H. Gregorio Marañón, and H. Fuenlabrada) that appeared during the first epidemic wave between March and June 2020. The patients included had to be receiving active cancer treatment, have a neutrophile count <1500/mL, have a fever (38 °C or more), and be infected with SARS-CoV-2 confirmed through a positive oropharingeal PCR. The anonymized data were collected in Excel 16.38. We included data on the variables related to demographics (age, sex, and hospital) as well as general clinical features: ECOG, performance status, smoking, body mass index, cardiovascular disease, or diabetes mellitus. We also collected information about the neoplastic disease (tumor location and stage), the anti-tumoral treatment (chemotherapy immunotherapy, targeted or endocrine treatments; time from last cycle; and COVID-19 diagnostic treatment intention), and multiple analytical parameters (neutrophil and lymphocyte counts, D-dimer, lactate dehydrogenase (LDH), pro-calcitonin, and C-reactive protein). Filgrastim^®^ administration (30–48 MU/0.5 mL subcutaneous per day) and treatment duration were also noted. Data related to SARS-CoV-2 infections were also included: severity and duration, as well as presence of pneumonia and/or thrombosis. Different COVID-19 treatments such as antibiotics, chloroquine, remdesivir, corticoids, anti-IL6 or anti-IL1, anticoagulants, and colchicine were collected, when applicable. Data about admission to the intensive care unit and the final outcome after SARS-CoV-2 infection were gathered: complete recovery, sequels, death, as well as cancer treatment modifications. This study was approved by the Ethical Committee for Clinical Research in La Paz University Hospital, code (PI-4194), on 4 May 2020. 

All of the statistical analyses were performed with the R 3.6.1 software [17]. Distributions of the quantitative variables were described through their mean, median, standard deviation, interquartile range, maximum, and minimum, and their normality was tested through the Shapiro–Wilk test. Distributions of qualitative variables were described though the corresponding absolute and relative frequencies. Bivariate analyses were performed for the categorical variables (e.g., related to basal factors, COVID-19 severity, etc.) vs. categorical (death, respiratory distress, etc.) or continuous-variables (e.g., lowest neutrophil levels, highest CRP, etc.). The significance of associations with the former was tested through Fisher’s exact test, while that with the latter was tested through one-way ANOVA or the Kruskal–Wallis test, depending on the normality of the quantitative variables. Multivariable models were derived in the form of logistic regressions to test the significance of the number of G-CSF treatment days in COVID19-related outcomes such as death or respiratory distress after adjusting for appropriate covariables unbalanced in the sample. All of the statistical tests were bilateral, and a significance level of 0.05 was used throughout. Statistical inferences in the logistic models used 95% profile confidence intervals. 

## 3. Results

Among the 943 patients with cancer and COVID-19, only 83 patients (11.3%) presented concomitant neutropenia during the first SARS-CoV-2 outbreak. 

### 3.1. Descriptive Analysis of Baseline Clinical Characteristics 

Eighty-three patients were selected from 14 hospitals in Spain. The baseline characteristics and different parameters related to cancer are presented in Table 1, both as totals and split by G-CSF treatment. The median age was 67 years, and the majority of patients presented ECOG 0-2 (79/95%). Cardiovascular disease was present in 35 (42.68%) patients, and 13 (15.8%) of total patients had diabetes. Weight data were also collected, observing 37/83 (54%) of patients with BMI > 25 (overweight), and the majority of patients were current smokers (17/83, 20.7%) or former smokers (31/83, 37.8%). On the other hand, the most frequent tumors, in decreasing order, were lung cancer (26%), breast cancer (22%), colon cancer (13%), and non-colon digestive cancers (17%), while the rest of the tumors included prostate cancer, ovarian cancer, gynecological cancer, sarcoma, and others. Advanced (IV) stage was the most common stage (67%), followed by stage III (19.5%) and stage II (13.4%). The most frequent cancer treatment was chemotherapy in 87.7% of the patients, followed by chemoimmunotherapy and other types in the rest of the participants. Finally, palliation was the main intention of the treatment (67.9%). Three patients who received a chemo-immunotherapy combination had a severe outcome and died. Table 1 also includes the *p*-value of the test for association of these variables with G-CSF treatment (Filgrastim^®^), from which we can see that the patients displayed similar clinical baseline features irrespective of receiving G-CSF treatment. The only exception was type of treatment, which is expected given that G-CSF treatment is more frequently administered in chemotherapy settings, even in a preventive way prior to cancer treatment (8 in our dataset). 

### 3.2. Descriptive Analysis of COVID-19 Disease and Treatments

The SARS-CoV-2 infection variables as well as treatments are presented in Table 2. Respiratory failure was detected in 63.4% of cases, and only 25,61% of patients did not require any oxygen supplementation. Nevertheless, 31 patients (37.8%) needed FiO_2_ supplementation of higher than 35%, whereas the rest of the patients required 24% oxygen flow or no extra oxygen flow. Fever or low-grade fever was referred to in 86.6% of the patients, while clinical and radiological pneumonia were confirmed in 63 patients (77%), being bilateral or multi-lobar in a considerable number of cases (45/83, 54.9%). The majority of the patients received antibiotics (77/83, 92.7%), mainly ceftriaxone and carbapenems. In addition to antibiotics, around 88% of the patients were treated with chloroquine, and antiviral therapy was used in 38 (46%) patients, predominantly lopinavir/ritonavir as first options (35%) followed by a combination of lopinavir/ritonavir and remdesivir (11%). Corticoids were administered in around 38% of the cases, 23% as high doses of methylprednisolone and 14% as standard doses. Other treatments were reported: anti-IL6 and colchicine, although with a very small amount of cases (14 and 1, respectively). The majority of patients (82%) received some kind of anti-thrombotic therapy, mainly prophylactic low-molecular weight heparin (68%), although only 12% of all patients had a (suspected or confirmed) thrombotic episode, and the mean highest D-dimer in this study was 1.558 ng/mL. Twenty-two patients (26.83%) met the criteria for intensive care. Eight of them died, but only one was admitted into intensive care. All patients who died had respiratory failure, and 46% of patients with respiratory failure eventually died. Oxygen requirements were also related to mortality, and 73% of patients with high oxygen flow died. Obviously, pneumonia severity was related to a higher risk of death as well as carbapenem treatment, being the most frequent treatment with antibiotics in those situations. The mean number of days of hospitalization was 12. In this series, 27 patients (30.2%) of all neutropenic oncological patients died after SARS-CoV-2 infection, while 69.7% of them were discharged after an improvement. Among the survivors (55 patients), 36 (68.5%) could continue their original treatment after the COVID-19 infection. We can see a trend towards a more severe COVID-19 infection in the G-CSF-treated patients, with higher proportions of severe pneumonia, thrombosis, days of hospitalization, and mortality. Especially striking is the highly significant increased proportion of patients with respiratory failure who require stronger oxygen support in the G-CSF group. In this group, only 13 patients survived and could retain their treatment prior to COVID-19 compared with the 24 no-filgrastim-treated patients (Table 2).

### 3.3. Descriptive Analysis of Neutropenia Characteristics and G-CSF Administration 

Table 3 includes the most relevant parameters collected about neutropenia and G-CSF treatment. All patients recruited had different grades of neutropenia during SARS-CoV-2 infection. Among the initial neutropenic patients, the lowest level of neutrophils reported was 0 cls/mm^3^. The mean was 707 cls/mm^3^, and the median was 650 cls/mm^3^. Around half of all patients (40, 49%) received growth colony stimulating factor (G-CSF) as routine dose of 5 ug/Kg to recover neutrophil counts. The rest of the patients in this series (43) did not receive this treatment for different reasons, mainly related to the uncertainty in its usefulness for patients with this type of cancer and infection with COVID-19: severity of neutropenia, doubts about lung inflammatory effect, and hospital protocols. The G-CSF treatment duration was very variable: from 1 day to 14, with a mean and median number of days on treatment of around 4.5 days. All patients who received G-CSF were on chemotherapy or chemo-immunotherapy treatment (see Table 1). In 26/40 patients (65%), G-CSF was initiated when neutropenia symptoms were detected but in the remaining 14/40 (35%), G-CSF was previously prescribed to prevent febrile neutropenia after a routine chemotherapy administration. After several days of G-CSF administration, the highest level of neutrophil count was 26.100 cls/mm^3^. Neutropenia outcome was variable: 21% had a worsening of neutrophil count and 50% improved. However, in 27% of the patients, their neutrophil count remained stable during SARS-CoV-2 infection. In addition, among the patients who were treated with G-CSF with a worsening of neutropenia, 91.6% died versus 0% among patients with total neutrophil recovery after G-CSF treatment. On the other hand, the mean lowest value of lymphocyte counts was 473/L, with no clear differences between G-CSF-treated patients vs. non-treated ones, while the mean of the highest LDH value was 685 U/L. The mean highest D-dimer value was 4513 ng/mL, that of calcitonin was 2.5 ng/dL, and that of C-reactive protein was 115 mg/dL. Again, we can see a trend in the data towards an increase in mortality according to the levels of these variables; in the case of D-dimer, pointing to an increase in thrombosis; while for LDH and the highly significantly increased C-reactive protein, signaling an increase in the number of inflammatory processes.

### 3.4. Bivariate Analyses of COVID-19 Infection in Patients with Neutropenia: Death Risk Factors 

Figure 1 shows a forest plot with the odds ratios (and corresponding 95% confidence intervals) for the association of different factors with mortality in our sample. In it, we could not find a significant association between age, ECOG, cardiovascular disease, type of tumor, or stage and mortality. Diabetes was related to a worse outcome and a higher risk of mortality compared with non-diabetes, although not significant (30.4% vs. 11.3%; *p* = 0.0533). The intention of treatment (palliative vs. curative) was also related to a trend of higher mortality in those patients with palliative treatment: 19 (83%) versus 4 (18%) patients, but this relation did not reach a statistically significant value (*p* = 0.064). Regarding oncological treatment, all patients treated with a combination of chemotherapy and immunotherapy died (3, 100%), but we could not find a significant association to a worse prognosis of SARS-CoV-2 infection and type of anti-cancer treatment received. Focusing on infections with COVID-19, in this series, we found that those patients with any type of pneumonia and neutropenia (unilateral, bilateral, unilobed, or multilobed) presented higher mortality compared with those without it: 19 (82.6%) versus 4 (17.3%), *p* = 0.0027. We also observed that men had more severe pneumonias compared with women, with higher oxygen requirements: 67.5% multilobe or bilateral in men versus 44.3% in women, *p* = 0.011. Although we did not find a significant relation between the type of tumor and COVID-19 mortality, we observed that 72.9% of lung cancer patients presented more severe infection with COVID-19, followed by colon cancer (41.2%), although this association was not significant (*p* = 0.064). We found an association between body mass index (BMI) and oxygen requirements when classified as no oxygen requirements vs. any oxygen flow. Oxygen support was needed in 57.2% of patients with BMI < 20, in 68% with BMI 20–25, in 82.6% with BMI 25–30, and in 100% with BMI > 30 (*p* = 0.0045). Smoking status was also associated with a higher risk for severe pneumonia and worse COVID-19 prognosis: 33 (73.3%) of smokers or former smokers developed some type of pneumonia vs. non-smokers, 12 (26.6%), *p* = 0.0035. In this series, 23 (27%) of all patients included died due to infections with COVID-19, and respiratory failure was present in all of them (100%). In our series, as we expected, higher oxygen needs were related to a significantly higher risk of mortality. We observed that low counts of neutrophils (*p* = 0.001) and lymphocytes (*p* = 0.013) were related to a higher mortality, and high values for LDH (*p* = 0.001) and Protein C (*p* = 0.003) were also associated with a worse outcome. In our cohort, corticoid administration in our patients with SARS-CoV2 infection and neutropenia was associated with a trend toward a lower mortality rate (52.17%) compared with patients without (67.9%) (*p* = 0.08). Regarding the impact of G-SCF administration, we observed a significant relation to mortality with 60.9% versus 39% in patients who did not receive G-CSF, as described before. There are also different conditions related to G-CSF administration, related as well to worse outcome, such as oxygen (*p* = 0.003) and carbapenems (*p* = 0.15) requirements (*p* = 0.003), pneumonia (*p* = 0.0585), and neutropenia severity (*p* = 0.035). Patients treated with G-CSF received corticoids more frequently compared with those without G-CSF (50% versus 16%). 

### 3.5. A Multivariable Model to Test the Effect of G-CSF on COVID-19 Severity 

We also carried out a multivariable analysis to test the potential effect of G-CSF treatment on COVID-19 mortality. In this analysis, we saw that the administration of G-CSF depends on multiple factors, with the hospital protocol being a very important one among them. For example, in the bivariate analyses, while tumor type; cancer treatment; treatment intention; ECOG; sex; age; and indeed, the lowest level of neutrophils, a proxy for neutropenia severity, are not associated with the hospital, as expected, the G-CSF treatment binary variable is *p* < 0.001. This fact suggests that patients with similar neutropenia conditions were treated in different ways depending on the oncologist criteria considering emergent data about the role of G-CSF in COVID-19 infection outcome. Thus, in some way, G-CSF treatment can be considered approximately randomized, and by adjusting for appropriate covariables, unbalanced in the sample and with a possible influence, the effect of G-CSF treatment on outcomes for these patients could be tested in order to gain insights about its possible beneficial or harmful effects. It would also be expected that the harmful or beneficial effects of G-CSF depend on the number of days it was administered, which has also a protocol-dependent component and is highly variable in this sample. Therefore, after removing a few patients for which G-CSF was administered in a preventive way, a logistic regression model was developed to predict respiratory distress as a function of the number of days of G-CSF treatment (Figure 2 and Figure 3). As adjusted covariates, sex, age, treatment purpose (palliative vs. curative, to adjust for global patient health status), tumor type, the lowest levels of lymphocytes (to adjust for immune status), and the lowest level of neutrophils in the patient (to adjust for neutropenic status) were used. A significant risk effect was obtained for the number of days of G-CSF treatment (OR = 1.40, 95% CI [1,2,05,07], *p*-value = 0.01). In Figure 2, we can see that the proportion of respiratory distress increases as we move from the first to the third intervals of treatment days. Superimposed is the fit of the logistic model. On the other hand, if death was used as a response variable instead, with the same adjusted variables, again, a significant risk effect was obtained for the number of days of G-CSF treatment (OR = 1.24, 95% CI [1,01,1,55], *p*-value = 0.04). Thus, a one-day increment of G-CSF treatment increases the odds ratio for respiratory distress by 1.4 and that of death by 1.24. This is represented in Figure 3, where the proportion of deaths is represented at three intervals for treatment days together with the fit of the logistic model. These results suggest that long neutropenic treatments in cancer patients could be harmful for the treatment of COVID-19 infection, instead of being beneficial.

## 4. Discussion

The COVID-19 pandemic has made daily oncological clinical care even more challenging. We performed a retrospective study of real-world data (RWD) of neutropenic oncological patients during the first wave of the COVID-19 pandemic in 14 hospitals in Spain. To our knowledge, this is the first analysis of SARS-CoV-2 infection in a significant number of cancer patients with neutropenia. We analyzed the impact of different variables in the SARS-CoV-2 infection outcome, mainly, G-CSF administration, which has been recently related to a probable worsening of SARS-CoV-2 prognosis [18]. Previous reports have reported around 13% of infections with COVID-19 in cancer patients [19], but in our study, this rate was 30.3%, probably due to the added risk of neutropenia, and it reached 60.9% in those treated with G-CSF. Some published studies have shown worse outcomes in patients with SARS-CoV-2 infection and neutropenia after treatment with G-CSF [18,20]. In our work, a direct correlation between G-CSF use and the severity of infection with COVID-19, respiratory failure, and death were found. Several putative confounding factors (sex, age, treatment intention, and lymphocyte and neutrophil counts) are possibly involved in the outcome, but a logistic regression model after adjusting for these covariables still seems significant, increasing the effect of the number of days of G-CSF treatment on death. Different immunological factors have been involved in severe coronavirus disease, associated with high levels of neutrophils and a high neutrophil–lymphocyte ratio (NLR) [4]. In our study, both neutrophilia and lymphocytopenia were correlated with a higher rate of respiratory failure and death, as it has been previously reported [21,22], specifically after G-CSF treatment [19,23]. In addition to the number of days of Filgrastim^®^ treatment, in our series, we found that higher levels of PCR, LDH, D-dimer, and lymphocytopenia have been related to poor prognosis in neutropenic patients with COVID-19, as has been previously described [24]. Before the COVID-19 pandemic, G-CSF use and neutrophil count recovery had been related with lung injury and Acute Respiratory Distress Syndrome (ARDS) [24,25]. In autopsy studies of patients infected with SARS-CoV-2, diffuse alveolar damage with hyaline membranes, hemorrhage, and neutrophilic infiltration have been reported [26,27,28]. Vascular neutrophilic inflammation and immune thrombosis are characteristics of COVID-19 infection [29] and have been described as being associated with G-CSF treatment. Neutrophil extracellular traps (NETS) may contribute to organ damage and mortality in COVID-19 disease. NETS are web-like structures of DNA and proteins expelled from the neutrophils that ensnare pathogens and have demonstrated a role in both venous and arterial thrombosis in several diseases [30] and can even promote cancer metastasis [31]. The intravascular aggregation of NETs in severe COVID-19 infection leads to immune thrombosis and disturbed microcirculation, organ damage, and ARDS [32,33]. Neutrophilia itself is associated with the release of cytokines, neutrophil activation, and NETs [34,35]. Increased NET formation correlates with ARDS, thrombosis, and cytokine storms, all of which are key in severe COVID-19 infections [36]. Filgrastim is a potent mobilizer of hematopoietic bone marrow cells and has a role activating cytokines to regulate T-cells and dendritic cell activation [37]. Some authors have suggested that we should be very cautious with G-CSF use in neutropenic patients with SARS-CoV-2 infection because there is a risk of triggering inflammation mediators related to severe COVID-19 infection [23,37]. We found a link between G-CSF and parameters of seriousness, a higher failure respiratory rate, higher neutrophil counts, lower lymphocytes, higher thrombosis, ARDS, and death. Although this is a small, non-randomized retrospective study and it was not designed to measure the effect of G-CSF treatment on the disease, a direct correlation of G-CSF treatment with a worse outcome was demonstrated after adjusting for possible confounding variables. This is a multicentric study of RWD describing neutropenic cancer management of patients with SARS-CoV-2 infection and showing a differential outcome for those treated or not with G-CSF under similar circumstances. Interestingly, corticoids have a beneficial effect in reducing mortality, as other authors have previously observed [38]. Morjaria et al. have published a pre-print study of 16 patients with neutropenia and COVID-19 treated with G-CSF. They classified these types of patients as good or poor responders according to the induced neutrophilia the day after G-CSF first dose administration. They found that good responders had a worse outcome, and they concluded that G-CSF treatment should be weighed in neutropenic patients with COVID-19 infections [20]. Moss et al. found that aged rhesus macaques infected by SARS-CoV-2 presented high G-CSF, involving the early release of neutrophils, less maturity, and less functionality compared with younger rhesus macaques, suggesting the potential role of GCSF administration in this mature human sample in worse COVID 19 outcomes [39]. Some authors have also described a link with neutrophil recovery and respiratory deterioration, and some reflections about this observed relation have been made in the literature [40]. According to the evidence as well as presented data, we should be more cautious in neutropenia management concurrent with infections by COVID-19 [41]. Nevertheless, other experiences suggest that G-CSF can counteract lymphocytopenia and could improve the outcome of the coronavirus disease [42]. A fully randomized clinical phase III trial could be designed to find the optimal balance, both in terms of the dose and duration of G-CSF treatment for these neutropenic cancer patients and with a more elaborate design (e.g., stratification) in terms of types of cancer. From this observational study, we have been able to obtain and test these findings about the risk of G-CSF treatment in patients with neutropenic cancer and infected by SARS-CoV-2 by applying appropriate adjustment variables, but further studies are required to confirm these findings and to obtain more information about aspects such as the involvement of cancer therapies, type of tumor, vaccination status, etc. 

## 5. Conclusions

In our retrospective study, we found a potential correlation between G-CSF treatment as well as the duration of G-CSF administration with the development of severe disease and mortality after adjusting for other potential variables that could also impact mortality in these types of patients.

## Figures and Tables

**Figure 1 cancers-13-04205-f001:**
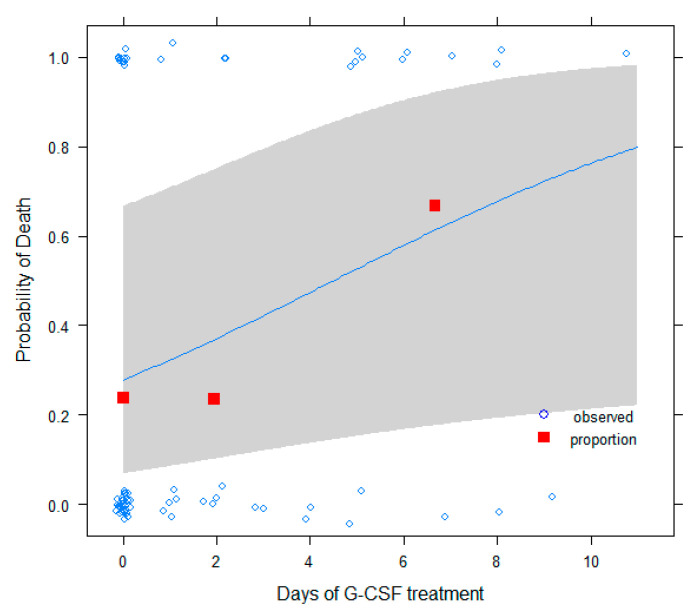
Death probability as a function of G-CSF treatment days: observed data (open circles, fitted to ease visualization), proportions (red squares, in subgroups of <1, 1–4, and >4 days), and fitted logistic model with 95% confidence bands are shown.

**Figure 2 cancers-13-04205-f002:**
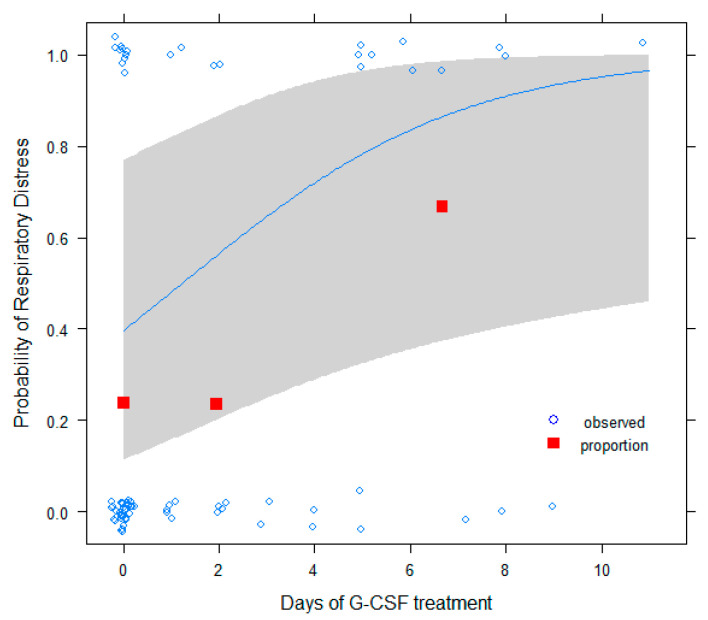
Respiratory distress as a function of G-CSF treatment days: observed data (open circles, fitted to ease visualization), proportions (red squares, in subgroups of <1, 1–4, and >4 days), and fitted logistic model with 95% confidence bands are shown.

**Figure 3 cancers-13-04205-f003:**
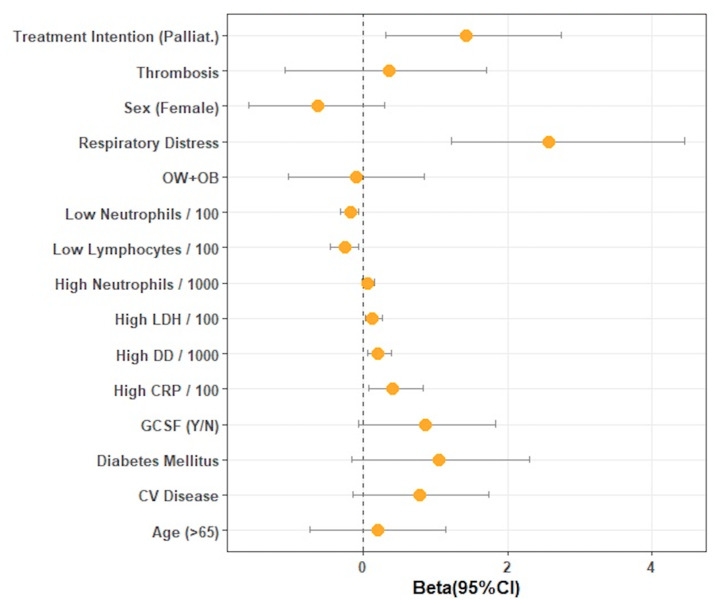
Forest plot showing mortality according to different variables related to neutropenia severity, treatment, and infection with COVID-19.

**Table 1 cancers-13-04205-t001:** Baseline characteristics of all patients included: total and split by G-CSF treatment. No attempt was made to impute missing values present in some variables. The *p*-value of association with G-CSF is also shown.

Baseline Characteristics	Total	Without G-CSF	With G-CSF	*p*-Value
Age				0.8
<70	36 (43.4%)	19 (44.2%)	17 (42.5%)	
>70	47 (56.6%)	24 (55.8%)	23 (57.5%)	
Sex				1
Male	41 (49.4%)	21 (48.8%)	20 (50%)	
Female	42 (50.6%)	22 (51.2%)	20 (50%)	
ECOG				0.15
0	24 (30%)	14 (34%)	10 (25.6%)	
1	46 (37.5%)	25 (61%)	21 (53.8%)	
2	9 (11.25%)	2 (4.9%)	7 (18%)	
3	1 (1.25%)	0	1 (2.6%)	
Cardiovascular disease *				1
No	47 (57.3%)	24 (57.14%)	23 (57.5%)	
Yes	35 (42.7%)	18 (42.9%)	17 (42.5%)	
Diabetes mellitus				0.37
No	69 (84.1%)	37 (88.1%)	32 (80%)	
Yes	13 (15.9%)	5 (11.9%)	8 (20%)	
Body Mass Index				0.44
<20	7 (10.1%)	5 (16.13%)	2 (5.26%)	
20–25	25 (36.2%)	9 (29%)	16 (42.11%)	
25–30	23 (33.3%)	11 (35.5%)	12 (31.6%)	
>30	14 (20.3%)	6 (19.3%)	8 (21%)	
Smoking				0.57
No	34 (41.5%)	17 (40.5%)	17 (42.5%)	
Smoker	17 (20.7%)	7 (16.7%)	10 (25%)	
Previous smoker	31 (37.8%)	18 (19.3%)	13 (32.5%)	
Primary tumor				0.66
Lung	22 (26.8%)	11 (26.2%)	11 (27.5%)	
Colorectal	11 (13.4%)	8 (19%)	3 (7.5%)	
Other digestive	14 (17.1%)	5 (11.9%)	9 (22.5%)	
Breast	18 (22%)	9 (21.4%)	9 (22.5%)	
Gynecological	6 (7.3%)	4 (9.5%)	2 (5%)	
Urothelial	3 (3.7%)	1 (2.4%)	2 (5%)	
Sarcoma	2 (2.4%)	0	2 (5%)	
Head and neck	5 (6.1%)	3 (7.1%)	2 (5%)	
Others	1 (1.22%)	1 (2.4%)	0	
Stage				0.88
II	11 (13.4%)	5 (11.9%)	6 (15%)	
III	16 (19.5%)	9 (21.4%)	7 (17.5%)	
IV	55 (67.1%)	28 (66.7%)	27 (67.5%)	
Type of treatment				0.001
Chemotherapy	63 (75.6%)	25 (59.5%)	38 (95%)	
Immunotherapy	1 (1.2%)	1 (2.4%)	0	
Chemoimmunotherapy	15 (12.2%)	13 (31%)	2 (5%)	
Other	3 (3.7%)	3 (7.1%)	0	
Treatment intention				1
Curative	27 (32.9%)	14 (33.3%)	13 (32.5%)	
Palliative	55 (67.1%)	28 (66.7%)	27 (67.5%)	

* Ischemic cardiac disease; history of high blood pressure; peripheral ischemic disease; other miocardiopathies.

**Table 2 cancers-13-04205-t002:** COVID-19 features of all patients included: total and split by G-CSF treatment. No attempt was made to impute missing values present in some variables. *p* value of association with G-CSF treatment is also shown.

COVID-19 Features	Total	Without G-CSF	With G-CSF	*p*-Value
Respiratory failure				<0.001
No	30 (36.6%)	23 (54.8%)	7 (17.5%)	
Yes	52 (63.4%)	19 (45.2%)	33 (82.5%)	
Oxygen support				0.002
No	21 (25.6%)	16 (38.1%)	5 (12.5%)	
<35%	30 (36.5%)	17 (40.5%)	13 (32.5%)	
35-50%	6 (7.3%)	0	6 (15%)	
>50%	25 (30.5%)	9 (21.4%)	16 (40%)	
Fever				1
No	11 (13.4%)	6 (14.3%)	5 (12.5%)	
Yes	71 (86.6%)	36 (85.7%)	35 (87.5%)	
Pneumonia				0.38
No	19 (23.2%)	12 (28.6%)	7 (17.5%)	
Unilobar	18 (22.4%)	10 (23.8%)	8 (20%)	
Multilobar/bilateral	45 (54.9%)	20 (47.6%)	25 (62.5%)	
Thrombosis				0.18
No	73 (87.9%)	40 (93.02%)	33 (82.5%)	
Suspicious/confirmed	10 (12.1%)	3 (6.98%)	7 (17.5%)	
In-hospital days (mean ± SD)	11.9 ± 9.6	10.6 ± 7	13.6 ± 11.6	0.09
Clinical evolution after neutropenia				0.45
Better/no changes	62 (78.4%)	34 (79.07%)	28 (70%)	
Worse	17 (21.5%)	9 (20.93%)	12 (30%)	
Death				0.1
Yes	27 (32.9%)	10 (23.8%)	17 (42.5%)	
No	55 (67.1%)	32 (76.19%)	26 (57.5%)	
Evolution after discharge *:				0.25
-No change of treatment	37 (68.5%)	24 (75%)	13 (59.1%)	
-Change of treatment	19 (31.5%)	8 (25%)	9 (40.9%)	

* Patient could maintain active treatment or required stopped active treatment and receive only definitive supportive management.

**Table 3 cancers-13-04205-t003:** Descriptive analysis of neutropenia variables.

Neutropenia and Inflammation Variables	Total	Without G-CSF	With G-CSF	*p*-Value
Lowest neutrophils count	616 ± 419	691 ± 369	541 ± 457	0.11
Highest neutrophils count	5423 ± 5078	3409 ± 2973	7538 ± 5940	<0.001
Lowest lymphocyte count	473 ± 310	467 ± 269	469 ± 356	0.77
Highest D-dymer	4513 ± 10134	2401 ± 3180	7456 ± 14880	0.062
Highest LDH	685 ± 1410	585 ± 604	792 ± 1943	0.86
Highest calcitonin	2.5 ± 5.3	2.6 ± 6.2	2.4 ± 3.6	0.06
Highest C-reactive protein	115 ± 143	90 ± 140	143 ± 143	0.001

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
