# Peer review of "A Multicenter Analysis of the Outcome of Cancer Patients with Neutropenia and COVID-19 Optionally Treated with Granulocyte-Colony Stimulating Factor (G-CSF): A Comparative Analysis"

_cancers, 2021, doi:10.3390/cancers13164205_

Round 1
Reviewer 1 Report
A comprehensive publication on the analysis of clinical features of cancer patients with febrile neutropenia and COVID-19 optionally treated with G-CSF.
Overall, the manuscript is worth publishing.
Please find below my comments:
- in the introduction, it is worth mentioning only the first author's surnames when citing their work.
- Literature for the sentence "Moreover, neutrophilia and high neutrophil/lymphocyte ratio (NLR) have been described as bad prognosis factors in COVID-19 patients [15]" should be added (https://doi.org/10.3390/pathogens9060493).
- The introduction could be cleaned up, especially the first part; it might be better to use shorter sentences and not give as many details.
- Would you please indicate the cities where the hospitals are located?
- The names of the software or drugs should include the manufacturer.
- At the beginning of the results, authors should define which types of cancer they included. This is very important to the readers.
- How was febrile neutropenia diagnosed?
- How G-CSF was used - doses, administration schedule.
- Table 3 is confusing - what do these mean, the highest and the lowest values?
- The discussion must be simplified.
- The conclusion is too general, and in light of so many results, it needs to be expanded.
- A native speaker must proofread the manuscript.
Reviewer 2 Report
In this small retrospective non randomized study, Sereno and colleagues found an association between G-CSF treatment and higher severity of COVID19 disease.
This analysis is clinically interesting and matches with the broad range of knowledge that G-CSF might increase probability/severity of cytokine storm, one of the main feature of COVID19- disease (PMID: 33399072).
However, it is still unclear from authors data whether benefits of G-CSF administration outweigh its risks.
Major points:
- The study suffer from the limited number of patients analyzed.
- The fact that patients are affected by different cancer forms and that they were treated with different pharmacological therapies complicate interpretation of author’s findings.
- Is Cancer diagnosis an independent risk factor for poor outcome?
- Could authors calculate p-values for each subcategory (Table I and II) ?
Minor Points:
- line 56 : Typo in SARS-COV2.
Round 2
Reviewer 1 Report
The authors have adressed all the comments of the reviewer.
Reviewer 2 Report
In this revised version authors did not add any information to the previous version and hence limiting the extent of their conclusions.
Considering the broad interest of the topic, I think that publish the manuscript in the present form will be in any case useful for the overall scientific community.